# The Analysis of the Morphology of the Saddle-Shaped Bronze Chips Briquettes Produced in the Roller Press

**DOI:** 10.3390/ma14061455

**Published:** 2021-03-16

**Authors:** Michał Bembenek, Janusz Krawczyk, Łukasz Frocisz, Tomasz Śleboda

**Affiliations:** 1Faculty of Mechanical Engineering and Robotics, AGH University of Science and Technology, A. Mickiewicza 30, 30-059 Kraków, Poland; 2Faculty of Metals Engineering and Industrial Computer Science, AGH University of Science and Technology, A. Mickiewicza 30, 30-059 Kraków, Poland; jkrawcz@agh.edu.pl (J.K.); lfrocisz@agh.edu.pl (Ł.F.); sleboda@agh.edu.pl (T.Ś.)

**Keywords:** roller press, copper alloy chips briquette, mechanical properties, structure, roughness

## Abstract

This paper presents the results of the investigations of the properties of saddle-shaped copper alloy chips briquettes produced in a roller press. The physical and mechanical properties of the investigated briquettes were examined on their external surfaces as well as on their cross-sections. The density, chemical composition, microstructure analysis obtained with a 3D and scanning microscope, surface roughness and hardness of the obtained briquettes were investigated. The research proved the differentiation of the physical and mechanical properties of briquettes depending on the investigated area of their surface. The analysis of changes in the porosity of briquettes on their cross-section showed zones of various densification levels. This research expands the knowledge of the processes taking place during the compaction and consolidation of granular materials in roller presses as well as the knowledge concerning designing the geometry of forming tools.

## 1. Introduction

The consolidation of fine-grained materials brings many positive economic, ecological and technological aspects [1,2,3,4,5]. Modern industry aims at the sustainable use of available raw materials and reduction of the production costs [6] while reducing the harmfulness of the process to the environment [7]. The aim is to reduce the energy consumption of the processes carried out [8] and to apply the remaining raw materials or waste for the production of the main product [9,10]. In the case of materials with a fine-grained structure, including chips, in order to reuse them, it is preferable to give them the appropriate geometry [11]. It should, first of all, allow for their easy and economical transport [12], storage or dosing to processing devices [13], as well as ensuring appropriate process management in processing units [14]. One of the materials that as a waste from the machining process can rationally be reused are copper and copper alloys. Waste chips based on copper or its alloys are often briquetted [15]. As a result of briquetting the material is properly packed [16], and the chips are joined together into a compact of the required density [17].

As a result of compacting the granular material, various types of bonds are formed between its grains [18,19]. This way, the compacts of a shape and size depending on the geometry of the working chamber of the device [20,21] and the methods of consolidation [22] are obtained, even several thousand times larger than a single grain [23]. The continuity of the briquetting process in roller briquetting machines allows for the reduction of energy consumption necessary to carry out this process [24]. Appropriate selection of the design parameters of this type of devices allows the achievement of a longer life of the forming tools as compared to other designing solutions for briquetting machines, such as screw briquetting machines [25,26,27]. High humidity, low bulk density, the presence of hydrophobic grains or high elastic deflection after removing the pressure may negatively affect the quality of the compacts produced in a roller presses, and in some cases even prevent it [20]. Materials with these physical and mechanical properties, such as copper chips, are described as difficult to briquette on roll presses [28]. As it results from our own research and the publications of other researchers, for this type of materials it is advisable to use an asymmetric compaction unit, in which the shape of opposite forming tool cavities is different [28,29,30]. This allows the elimination of the phenomenon of briquette cracking in the plane of the contact of tool cavities of both rollers, in so-called dividing plane [31]. Such an effect is achieved due to a more favorable distribution of pressures acting on the briquette during formation in the die cavity as compared to the symmetrical layouts [30]. An important aspect influencing the process of proper material forming in the forming cavities is the composition of the consolidated material [32]. It is also known that, apart from the binder [33,34], some materials added to the basic mixture may improve its susceptibility to consolidation, which results in obtaining briquettes with better usable parameters, such as hardness and strength [35,36,37]. The proper composition of the mixture subjected to consolidation is also important not only from the point of view of obtaining high mechanical strength of briquettes, but also the correctness of the subsequent process with the use of the produced briquettes [38]. Therefore, part of the research is also focused on the optimization of the composition of agglomerates and the parameters of the consolidation process in relation to the operations in which the briquettes are used [39,40]. Additions introduced at the stage of the briquette consolidation act, inter alia, as iron oxides reducing agents in steelmaking processes [41,42] leading to reduction of slag foaming [43].

The assessment of the morphological features and the mechanical properties of briquettes provides a lot of information on the correctness of the selected parameters of the consolidation process, the correctness of the geometry of the forming cavities and the properties of the mixture, and also allows the development or verification of the adopted methods of process simulation [44,45,46]. Determining the relationships between the properties of the integrated material, the method of carrying out the consolidation process and the features of the obtained product allows control over the most important parameters of the process in order to obtain a product with the required final parameters [47]. The characteristics of the obtained product are assessed on the basis of a number of tests, often differentiated in terms of the methods used. The conducted research allows determination of the tendencies and predisposition of the material to specific characteristic behaviors occurring during consolidation and influencing the obtained product. On the basis of this type of research, it was observed that in the case of many materials, the compressive strength of the finished briquettes increases with the increase of surface roughness average R_a_ [48]. This phenomenon is related to the lower susceptibility to plastic deformation of the higher strength particles included in the briquetted material. This information was obtained by comparing the results of the roughness and hardness tests of briquettes. It is important in this case to emphasize that these results refer only to the central part of the volume of the briquettes. However, they do not provide the information on the relationship between roughness and strength in other characteristic areas on the briquette surface. Moreover, usually the distribution of the morphological features of the briquette in its entire volume is ignored and is limited to the visual assessment of the briquette [49]. The information on the briquette external surfaces morphology can be very useful for the correct modeling or correction of forming tools cavities. Studies available in the literature describe mainly the compressive strength of briquettes in relation to their density [50], however, the influence of the quality of the obtained briquette surface on the local briquette density and its distribution has not been investigated so far. Therefore, the aim of this study was to obtain the information on the physical and mechanical properties of the external surfaces and the cross-sections of the investigated saddle-shaped briquettes produced in a roller press. In order to obtain the maximum amount of information, chips from the machining of copper alloys were selected as the experimental material.

## 2. Materials and Methods

The first stage of the tests was to consolidate the materials to produce briquettes (Figure 1). This was done using a roller press with a 450 mm roll pitch diameter with an installed compaction unit for producing saddle-shaped briquettes (Figure 2) with a rated capacity of 6.5 cm^3^ with a size of 31 mm × 30 mm × 13 mm. The outline view of the tools cavity used to consolidate the material is presented in Figure 3. The press was equipped with a cycloidal gear motor of a power of 22 kW and a frequency converter enabling infinitely variable control of the speed of the rolls. All materials were consolidated using a gravity feeder with a roller speed of 4.25 rpm, which corresponded to the peripheral speed of the rolls equal to 0.1 m/s with an inter-roll gap of 1 mm.

In order to identify the surface and shape of a single briquette, two ends of the sample were marked according to how the briquette comes out of the briquetting machine: top (side with well compacted material) and bottom (side of the briquette with insufficient degree of compaction). The tests were carried out on two surfaces of the samples: front (surface from the groove of the forming roll) and back (surface from the forming cavity on the forming roll). Each surface was marked with two colors for better identification (Figure 4). The colors change where the plane is symmetrical. To measure the chemical composition, observe the microstructure and measure the hardness and roughness, the briquettes were outlined with a mesh on both surfaces according to the diagrams shown in Figure 5 and Figure 6. Grid lines were marked every 5 mm. Columns and rows were described with the first letter from the name of the side: F—front, B—back, and subsequent letters of the alphabet and numbers.

In order to determine the chemical composition of the investigated briquette, a metallographic cross-section was made along the edge 2. A scanning microscope (Japan Electron Optics Laboratory Co., Ltd., Tokyo, Japan) was used for the research. Sample for the microscopic analysis was prepared by the gridding on the SiC sandpapers with gradation from 120 to 2400. Analysis of the material microstructure was performed at the center area of the sample height. After gridding, the sample was polished with the use of synthetic diamond with the gradation from 3 to 0.24 μm. The obtained metallographic cross-section was also used to carry out the EDS analysis (Energy Dispersive Spectroscopy), which allowed for the identification of the elements included in the investigated material (FOV: 134 µm, voltage: 20 kV, current: 10 nA, analysis time: 60 s). A scanning microscope (JEOL, Ltd., Tokyo, Japan) was also used to analyze the microstructure on a representative area of the sample. The phases were defined by determining the chemical composition of their presence in the analyzed areas. The Archimedes method was used to determine the briquette density using the Equation (1).
(1)$$\;\rho =A\cdot \rho_{c}\;$$
where ρ —density; A —dimensionless quantity, defining the ratio of the density of the tested body to the density of the liquid; and $\rho_{c}$—density of the investigated briquette.

The dimensionless quantity A was determined from the Equation (2):(2)$$\;A=\frac{\rho }{\rho_{c}}=\frac{Q}{Q-Q_{1}}=\frac{mg}{mg-mg_{1}}=\frac{m}{m-m_{1}}$$
where *Q*—material weight in the air; *Q*_1_—material weight in water; *m*_1_—apparent value of the material weight after full immersion in the liquid (g); *m*—material weight in the air (g); and *g*—acceleration due to gravity (m/s^2^).

To determinate the briquette hardness, measurements were carried out based on the Leeb dynamic method. An Equotip Bambino 2 (Proceq AG, Zurich, Switzerland) hardness measurement device with a type D universal head was used, and the test was repeated four times for each of the investigated materials in the central part of the briquette surface. During the test, the sample was fixed in the holder, so that it was stable during the measurement. The roughness measurements were performed using a Veeco WykoNT9300 optical profilometer (VEECO, Los Angeles, CA, USA). Roughness average (R_a_), maximum height of the profile (R_t_), and root mean square roughness (R_q_) measurements were made on the back surface in the areas BA4 to BF4, while on the front surface in the areas FF1 to FF4 (Figure 7). Roughness was measured for the areas of 1.8 mm × 2.3 mm. The longer side was arranged perpendicular to edge 1 or edge 2, depending on the investigated briquette plane, while the shorter side was parallel to it. The briquette curvature error was considered as a systematic error.

Macroscopic measurements at 100× magnification were made with a Keyence VHX-7000 microscope (Keyence, Osaka, Japan) along the edge 2 on the back and edge 1 surfaces on the briquette front surface. Regions BA4 to BF4 on the back surface and FF1 to FF6 on the front surface were observed. Observations were made on approximately 2.3 mm × 3.0 mm slices. The cuttings were located in the central part of a given area. The longer edge of the slice was aligned parallel to edge 1 or edge 2 depending on the plane of the briquette being investigated.

The internal structure of the briquette was observed on a specimen made along the edge 1 on the back surface. The sample was prepared by the same technique as in the case of SEM analysis. Then the mesh was applied to the plane of intersection. The grid lines were arranged every 5 mm, the same as for the grid on external surfaces. The grid superimposed on the intersection surface is illustrated in Figure 8. Measurements were made on the intersection plane surface with the measurement areas from W1 to Y4.

Using the mesh superimposed on the cross-section of the briquette, documentation of the briquette porosity measurements was made. The porosity values were expressed as the percentage of pores on the specimen surface. They were presented as a map of the porosity distribution on the material cross-section. The individual values of the porosity were determined as the average value of the porosity from four areas of a particular area on the cross-section. The porosity was measured by computer image analysis method using the Metlilo 9.11 software (9.11, Janusz Szala, Cracow, Poland). The porosity value was defined as the percentage of dark areas to the total area of the observed area of the briquette cross-section. Based on this dependence, the stereological parameter V_v_ (relative phase volume) was determined. In the case of the investigated material, it was the relative volume of the porosity in a given area of the investigated sample.

## 3. Results and Discussion

Based on macroscopic observations of consolidated briquettes, it can be concluded that the briquettes are characterized by a higher degree of compaction in the upper side, while the material has not been properly compacted in the lower part. The bottom side of the briquettes is delicate and easily crumbles. A representative area of the specimen cross-section was used for the analysis of the chemical composition of the obtained briquettes (Figure 9).

After the analysis of the X-ray diffraction pattern (Figure 10), the chemical composition was assessed, which is presented in Table 1. In the spectrum analysis, the low-energy components of oxygen and carbon were removed. These elements were not analyzed due to the imperfection of the EDS method in their assessment.

The obtained results suggest that the investigated material was a briquette made of lead bronze chips, which also contained tin and traces of impurities in the form of silicon and iron. The reported iron content could be added to improve the mechanical properties of the bearing alloy [51]. Both very high lead content as well as the addition of Fe can be a residue from the machining process. The addition of lead to copper allows easier chip formation and material fragmentation along the lead precipitations in the material. In addition, the application of high loads causes “squeezing out” of lead onto the surface of the chip. The presented effects may significantly affect the obtained results [52]. Cu-Pb-Sn materials are used for bearings, pump elements and soft bushings. Lead, as already mentioned, due to its properties helps in machining and prevents the bearings from seizing up. Due to the toxicity of lead, briquetting of this material avoids its dusting.

In the next part of the research, the microstructure of the briquette in the area of its core was analyzed, a representative area being shown in Figure 11. There are visible components of the microstructure in the form of chip cross-sections with their porosities. Figure 12a–g shows the briquette microstructure observed under a scanning electron microscope.

By analyzing the observed microstructure, it was possible to notice that the chips are clearly two-phase in nature (Figure 12a–c). The areas facilitating their connection were also observed. The morphology of the chip particles indicated that their shape corresponded to the so-called short step chips. The observation of the chip microstructure at higher magnification (Figure 12f,g) allowed for the observation that the initial material was composed of a two-phase structure. A significant amount of the light phase in the vicinity of the darker matrix was observed. The matrix most likely consists in this case of a solution of tin and iron in copper, while the lighter areas are the precipitations of almost pure lead due to its low solubility in copper. The observed precipitations have a clearly elongated shape, which is related to chip deformation during the briquetting process (Figure 12d,e) [51,53].

After the analysis of light and dark precipitates (Figure 13, Table 2) it was found that the light areas correspond to the areas of lead particles, while the gray areas characterize the phase, that is, the solid solution of tin in copper. The microstructure morphology confirmed that the lead precipitates facilitated chip fragmentation and facilitated briquette particles consolidation. Due to its low melting point, lead significantly facilitated the joining of particles under pressure. The addition of lead significantly improves the compactibility of copper powders during sintering [54]. In the case of its significantly higher content during the briquetting process, it creates bridges that connect loose particles (Figure 12d–g), facilitating the consolidation of chips.

Table 3 shows the results of briquette weight measurements. Based on the obtained results, the briquette density, which was 6.91 g/cm^3^, was determined in accordance with the Archimedes method. 

The results of the hardness measurements are summarized in Table 4 and Table 5. The results of the hardness measurements are presented in Figure 14 and Figure 15.

In most of the cases, the average hardness of the briquette outer edges was much lower. By analyzing the back surface, it was noticed that the greater hardness occurred closer to the top of the briquette, that is, the area where the briquette was better consolidated. In the case of the front surface, the hardness in both the upper and lower briquette areas were differentiated and reached similar small values. The highest hardness values on this side of the briquette were observed in its central area.

The examples of the measurement results of the structure and surface roughness in a form of the surface development for the BD4 area on the back surface and the FF3 area on the front surface are presented in Figure 16. Table 6 and Table 7 summarize the collective roughness measurements results for the tested areas.

Based on the obtained results it was possible to notice that the roughness changed in a narrow range on the back surface along edge 2. The roughness parameters were symmetrically distributed in relation to the central areas and decreased as the areas moved closer to the briquette edges (Figure 17 and Figure 18). Small changes in the roughness average R_a_ value in the considered areas may indicate the correct compaction of the material along edge 2 on the back surface. In the case of the front surface, the greatest roughness occurred near the bottom of the briquette, that is, the area where the briquette could easily crumble. A significant increase in the roughness parameters was visible along with the movement along edge 1 from the top of the briquette to its bottom (Figure 19 and Figure 20).

The results of the measurements of hardness and roughness of the surface layer show only a partial correlation. This is due to many factors and also because the hardness was assessed by the Leeb dynamic method and then converted. This methodology was adopted due to the characteristics of the material in terms of the cohesion and geometry. Therefore, an increase in roughness in this case may act on the resilience (which was basically measured). The material under the surface layer, especially the degree of its compaction, also plays an important role. Conceptual work on the appropriate methodology for measuring hardness in the briquette cross-section is currently underway.

Observing the macroscopic pictures of the briquette on the back surface (Figure 21) taken along edge 2 it was possible to notice that the arrangement of the chips and the degree of their deformation were similar along the entire length of the ridge. Only in the areas of BA4 and BF4 (i.e., areas located closer to the edge) was better ductility of the consolidated material visible.

In the case of the observations made on the surface of the front of the briquette along the edge 1 (Figure 22), good ductility of the chips near the top of the briquette was visible. It was noticed that as the chips moved towards the bottom of the briquette, the chips became more visible. This, in turn, led to the conclusion that the briquettes were poorly consolidated in the vicinity of the briquette bottom.

The results of the porosity of the investigated areas on the briquette cross-section are shown in Table 8, while the map of the porosity distribution on the briquette cross-section is shown in Figure 23. The examples of pictures used to measure porosity are shown in Figure 24. Figure 24a,d shows large deformations of the chips (briquette top part), which are smaller towards the briquette center (Figure 24b,e,g). The smallest chip deformation is observed in the back and bottom zones of the briquette (Figure 24c,f,h,i).

The performed porosity analysis allowed the observation that the highest densification was observed in the upper area of the briquette. The porosity in these areas of the briquette differed by 13–16%. With the distance from the upper part of the briquette, a gradual increase in its porosity was observed. Much greater porosity was visible in the areas X3 to X6. The highest values of the porosity were observed in the final region of the sample, where the porosity in the X6 region increased above 50%. Therefore, chipping the particles of the lower part of the briquette was observed.

## 4. Conclusions

The conducted investigations concerning saddle-shaped copper alloy chips briquettes provide a lot of important information on the consolidation of copper chips with the use of roller presses with an asymmetrical compaction system. On the basis of this research, it was possible to accurately determine the chemical composition of the alloy from which the chips were obtained as well as to determine the type and the mechanism of their breaking during cutting operations. The lead, as one of the investigated alloy constituents, acted as a binder and facilitated the consolidation process. The hardness tests showed that the briquette had the highest hardness on the top surface and on the center of both investigated surfaces. The hardness at the edges of the briquette was up to approximately 2.5 times lower than on its central and upper part. For both the front and back surfaces, the highest surface roughness is obtained in the central part of the briquette. The results of the porosity tests also showed a better internal densification of the top part of the briquette. The curves of the porosity distribution in the cross-section of the investigated briquette indicated a lower porosity of the material on the front side of the briquette, which coincided with larger areas of higher hardness and lower roughness on the front surface. The research is innovative and shows the method of compacting the metallic material on a roller press. The research shows the direction of improving the saddle-shaped cavities (shallow the lower part) in order to obtain from copper alloy chips good quality briquettes with homogeneous parameters in the entire volume of the briquette. Degree of compaction of the material under the surface layer as well as interaction between briquette core and its surface play important roles and will be investigated in the next stages of this research. Conceptual work on the appropriate methodology for measuring hardness in the briquette cross-section is currently underway.

## Figures and Tables

**Figure 1 materials-14-01455-f001:**
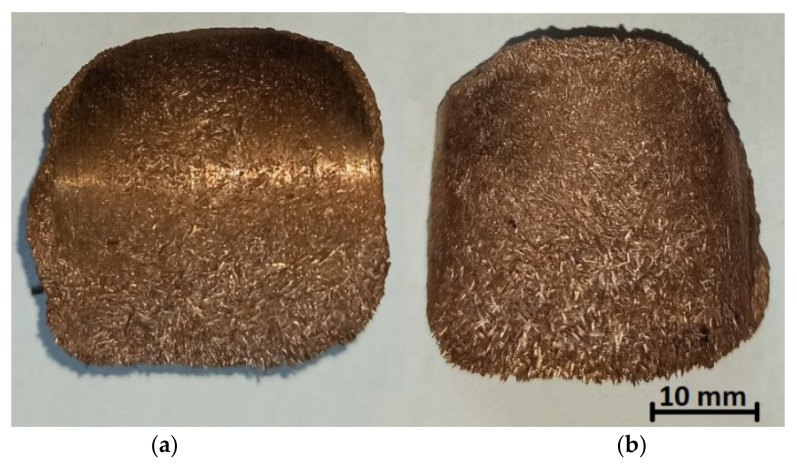
Briquettes used for testing: (**a**) back, (**b**) front.

**Figure 2 materials-14-01455-f002:**
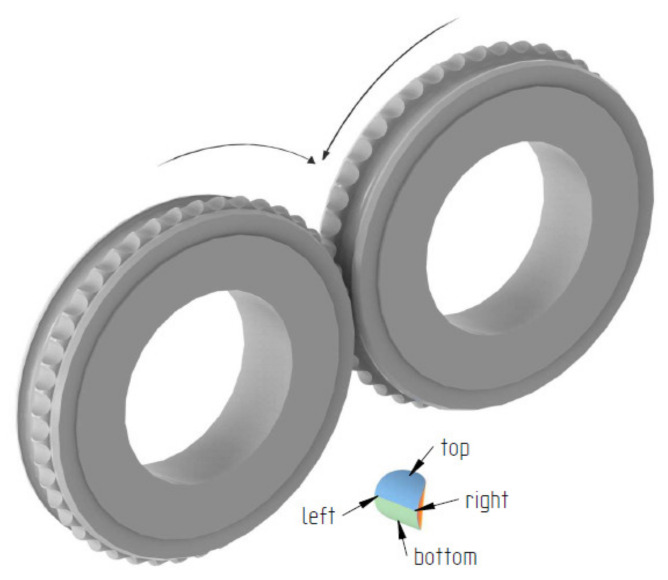
Forming rings used in roller press with marked rotation direction.

**Figure 3 materials-14-01455-f003:**
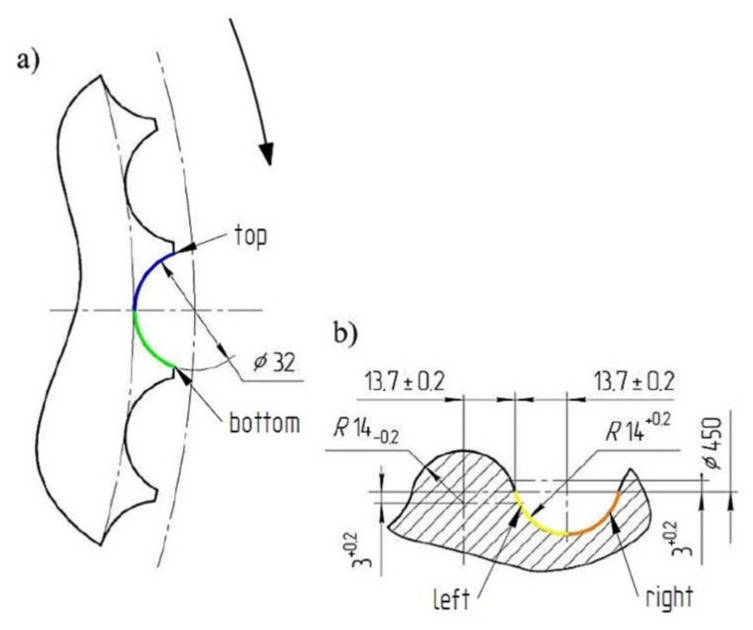
Geometry of tools cavities on the working surface of rolls used for briquetting: (**a**) front view, (**b**) cross section through the groove.

**Figure 4 materials-14-01455-f004:**
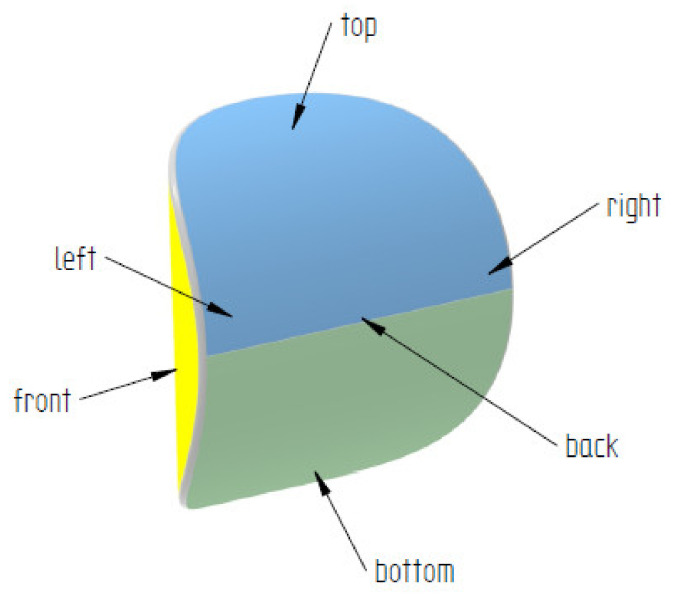
Saddle-shaped briquette with selected markings.

**Figure 5 materials-14-01455-f005:**
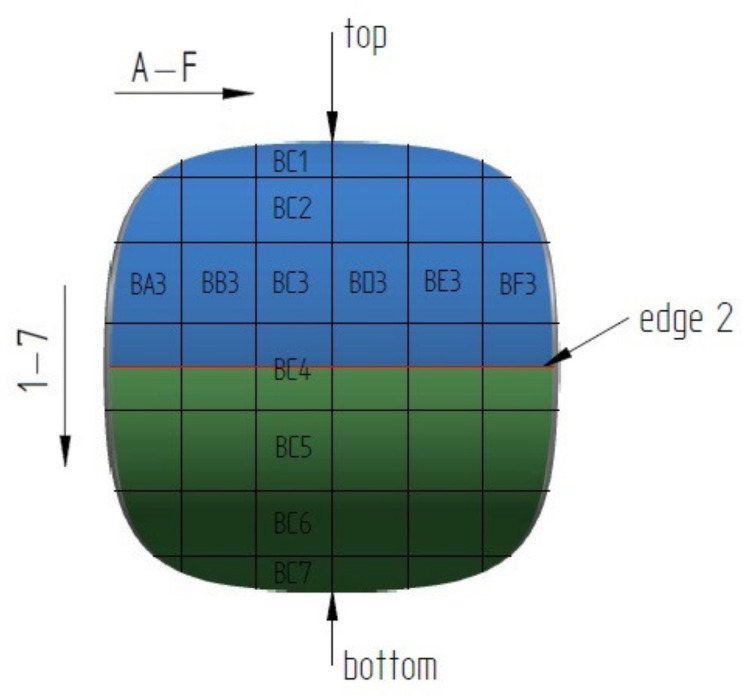
Back surface with the way of notification of designated areas.

**Figure 6 materials-14-01455-f006:**
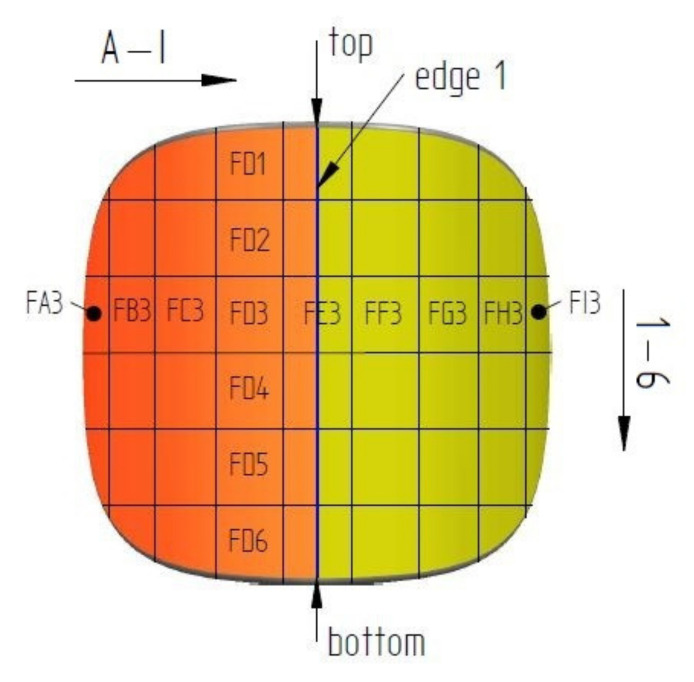
Front surface with the way of notification of designated areas.

**Figure 7 materials-14-01455-f007:**
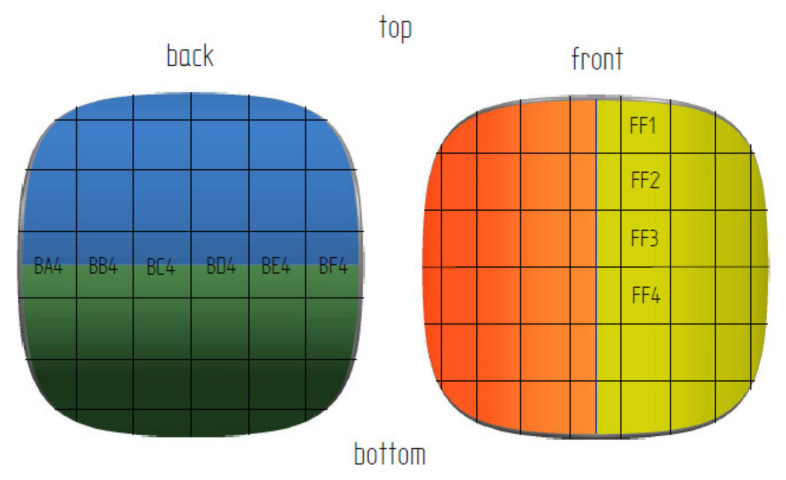
Briquette with marked areas for roughness measurements.

**Figure 8 materials-14-01455-f008:**
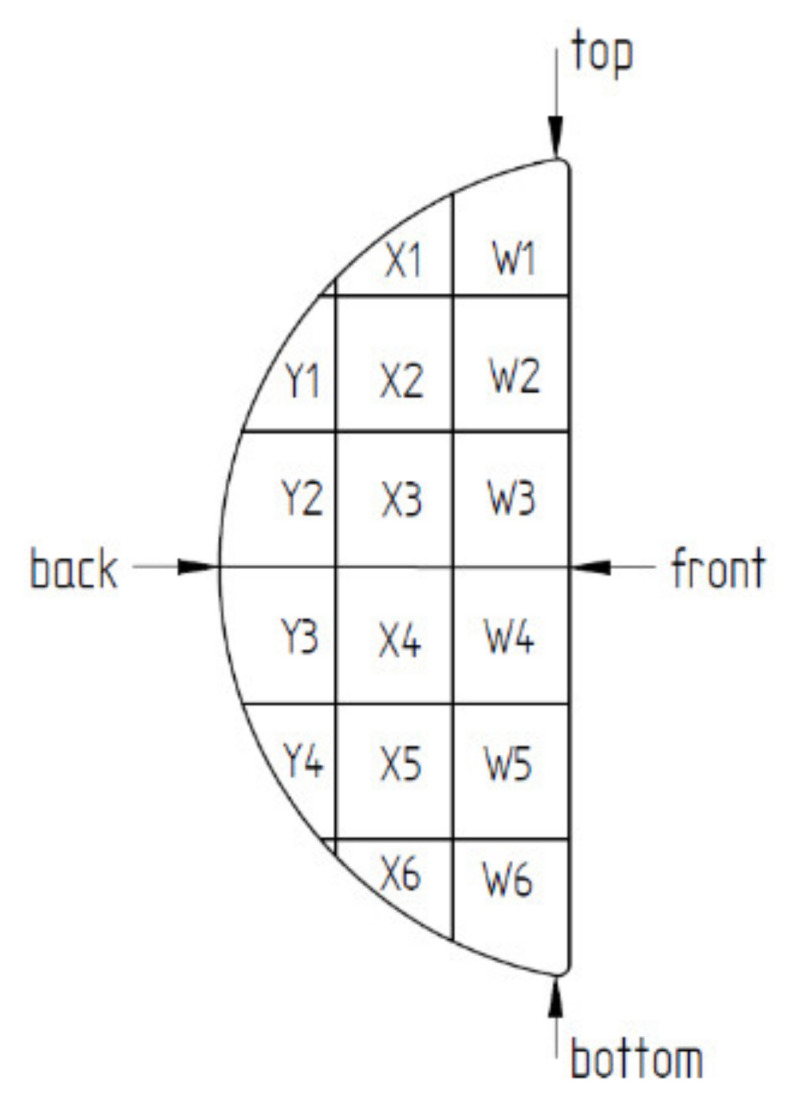
Cross-section of the briquette with the grid fields marked.

**Figure 9 materials-14-01455-f009:**
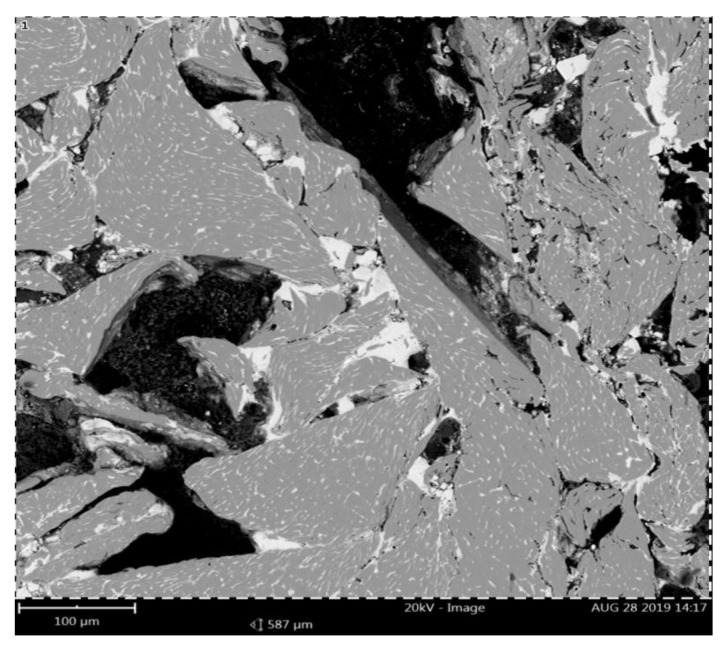
Representative microstructure area for EDS analysis.

**Figure 10 materials-14-01455-f010:**
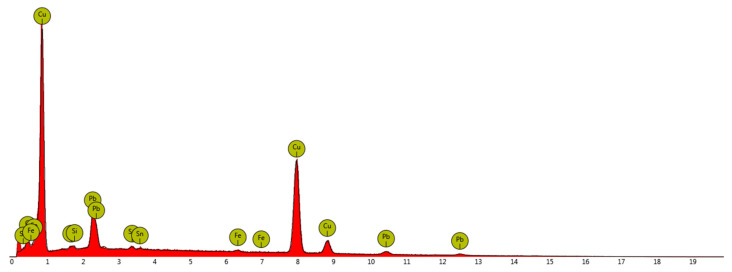
EDS spectrum for a representative sample area.

**Figure 11 materials-14-01455-f011:**
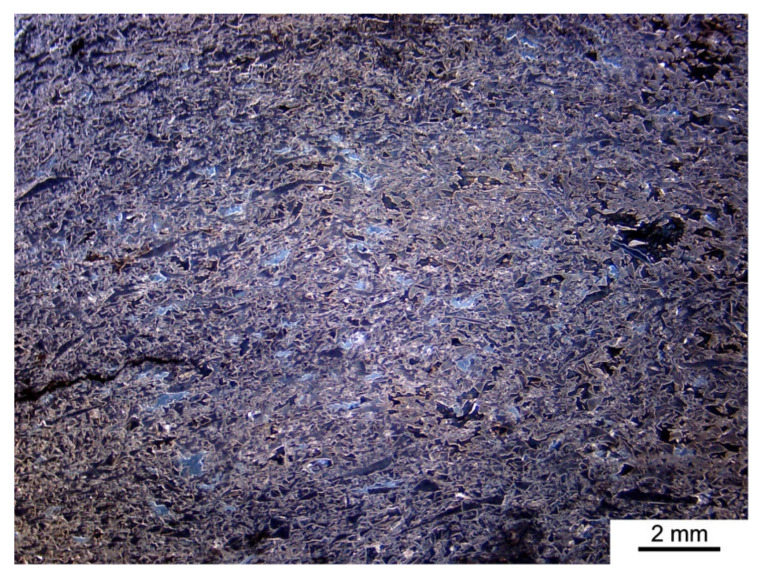
A representative area of the sample (10×).

**Figure 12 materials-14-01455-f012:**
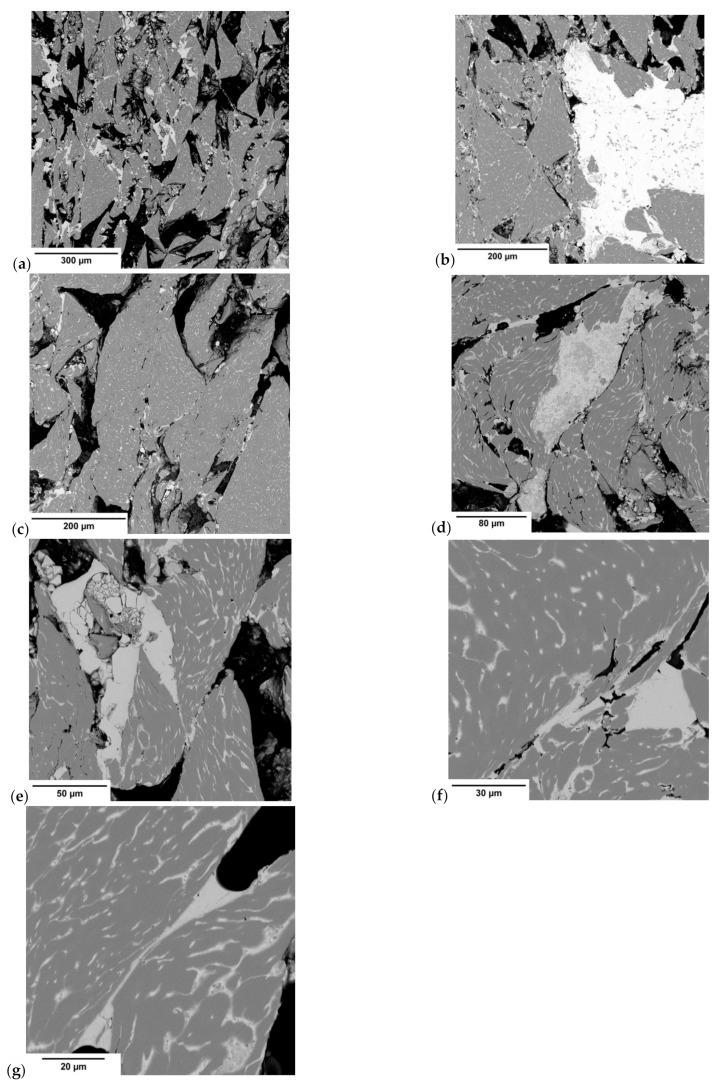
Briquette microstructure observed under a scanning electron microscope (SEM): (**a**–**g**) individual images.

**Figure 13 materials-14-01455-f013:**
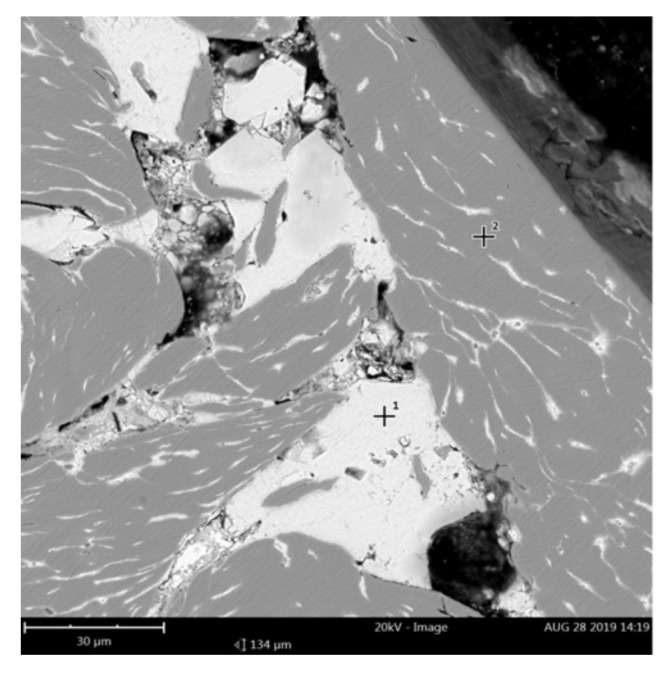
Microanalysis in the light and gray phase area—analysis site marked with a cross.

**Figure 14 materials-14-01455-f014:**
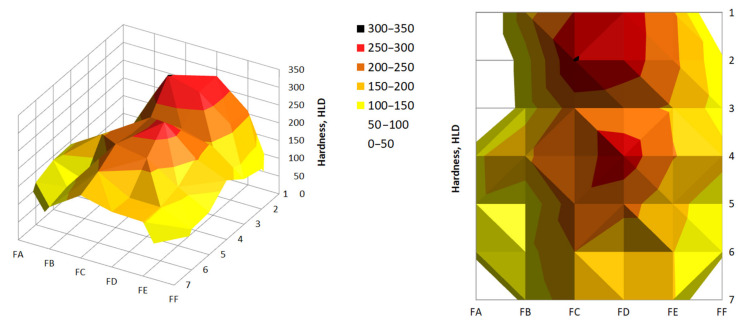
Hardness distribution on the back surface of the briquette.

**Figure 15 materials-14-01455-f015:**
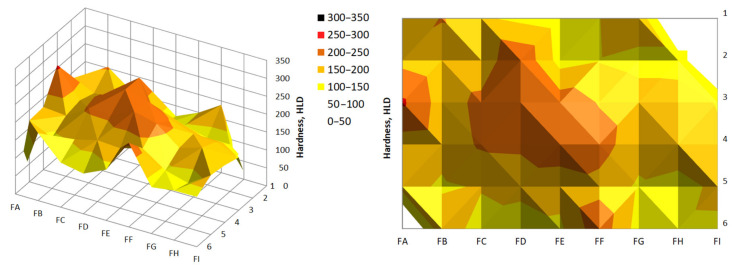
Hardness distribution on the front surface of the briquette.

**Figure 16 materials-14-01455-f016:**
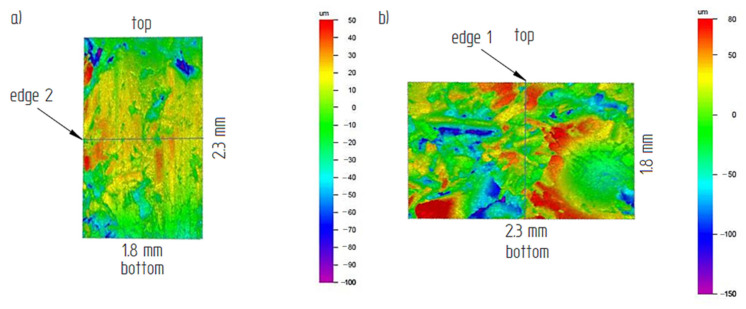
Example of surface roughness measurements area: (**a**) BD4 on the back surface; (**b**) FF3 on the front surface.

**Figure 17 materials-14-01455-f017:**
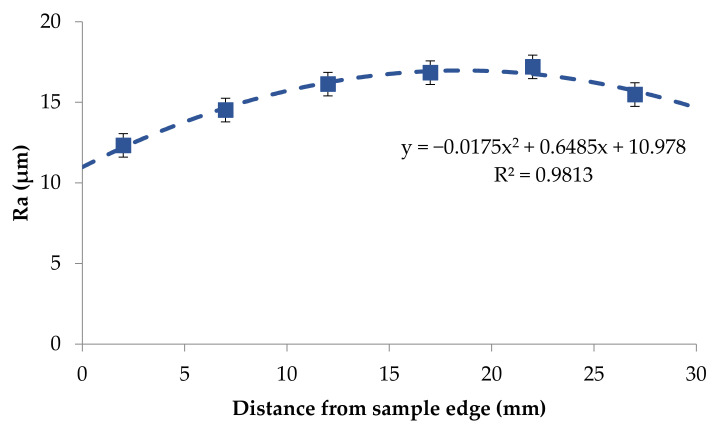
The relationship between the roughness average R_a_ and the distance on the edge 2 on the briquette back surface—left to right direction.

**Figure 18 materials-14-01455-f018:**
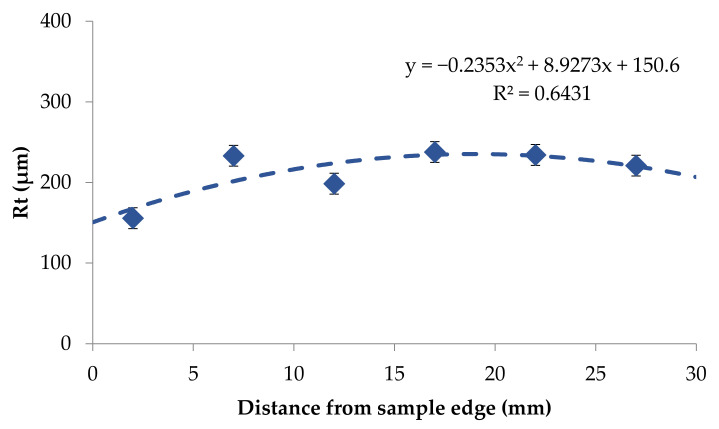
The relationship between the maximum height of the profile (R_t_) and the distance on the edge 2 on the briquette back surface—left to right direction.

**Figure 19 materials-14-01455-f019:**
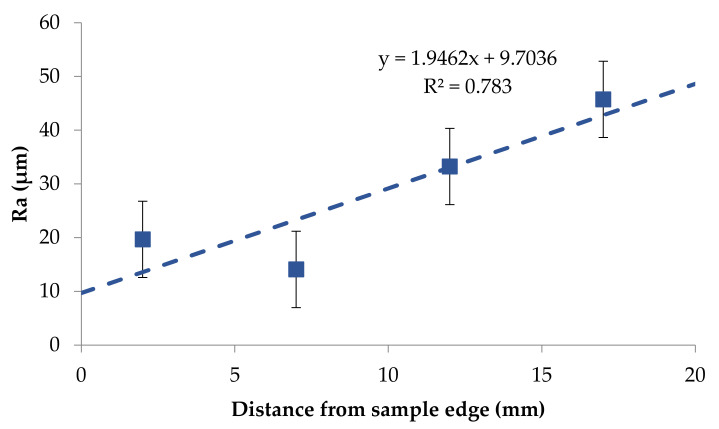
The relationship between the roughness average R_a_ and the distance on the edge 1 on the briquette front surface—top to bottom direction.

**Figure 20 materials-14-01455-f020:**
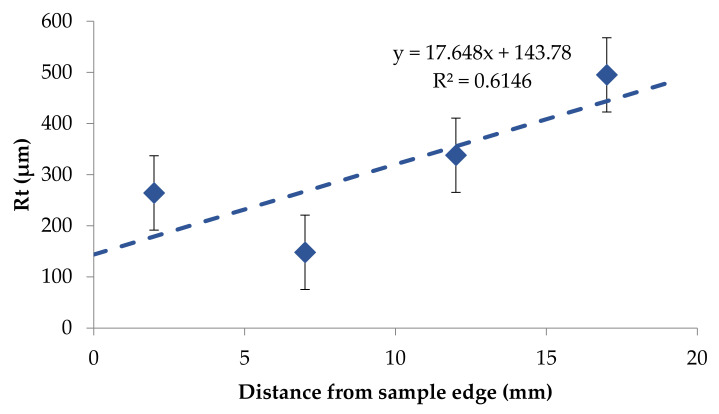
The relationship between the maximum height of the profile (R_t_) and the distance on the edge on the briquette front surface—top to bottom direction.

**Figure 21 materials-14-01455-f021:**
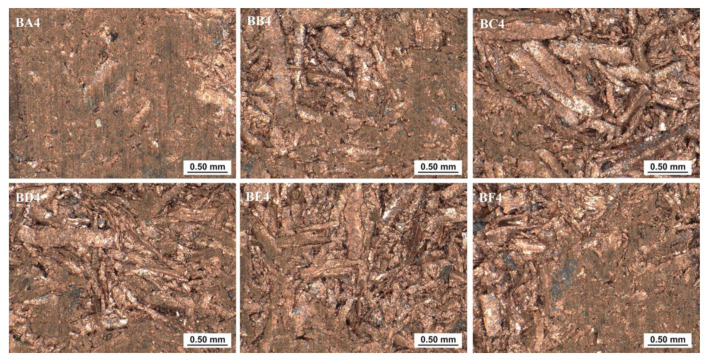
Pictures of the selected areas of the back surface of the briquette.

**Figure 22 materials-14-01455-f022:**
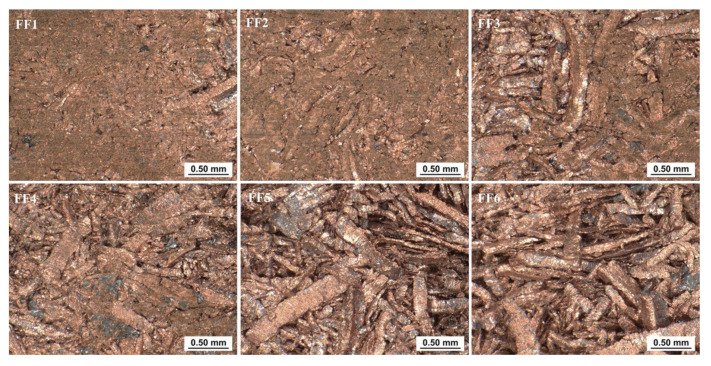
Pictures of the selected areas of the front surface of the briquette.

**Figure 23 materials-14-01455-f023:**
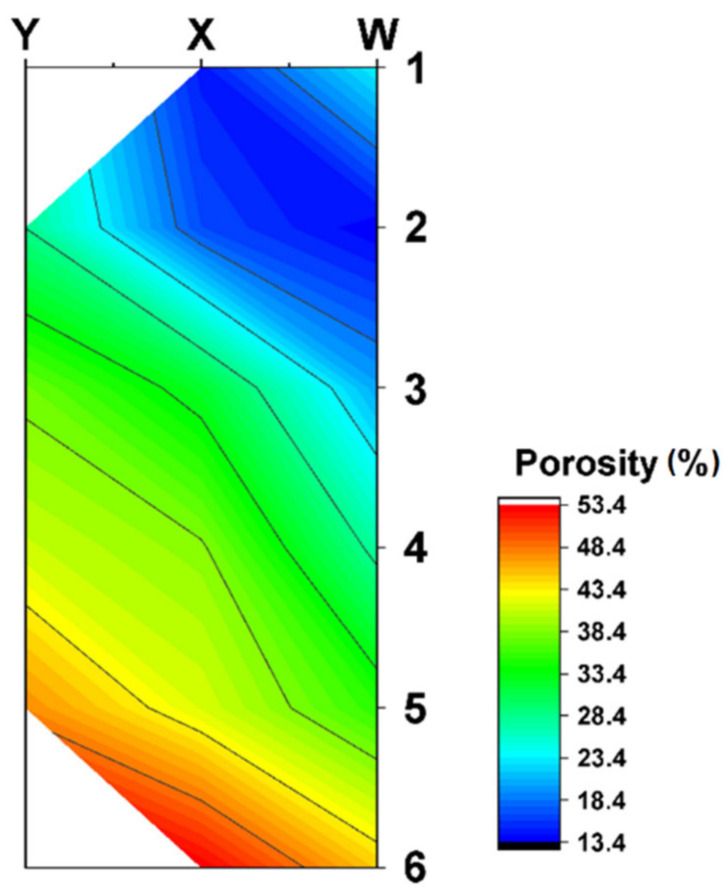
Map of the porosity distribution on the briquette cross-section, respectively for each of the briquette cross-section areas.

**Figure 24 materials-14-01455-f024:**
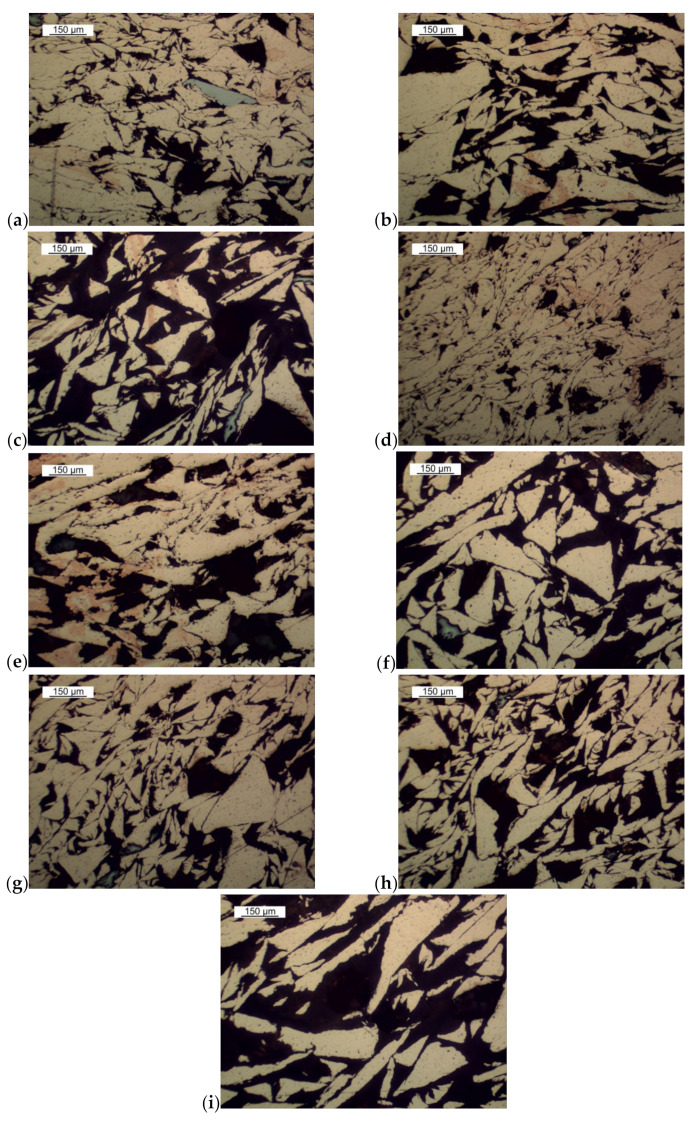
Microstructures of the selected areas of the cross-sections of the briquette: (**a**) W1, (**b**) W3, (**c**) W5, (**d**) X1, (**e**) X3, (**f**) X5, (**g**) Y1, (**h**) Y3, (**i**) Y4.

**Table 1 materials-14-01455-t001:** Chemical composition of the tested material.

Element	Cu	Pb	Sn	Fe	Si
Weight %	51.7	46.0	1.5	0.5	0.3

**Table 2 materials-14-01455-t002:** The results of the microanalysis in the light and gray phase.

Point		Pb	O	Cu	Si	Sn
Weight %	1	94.3	4.9	0.6	0.2	–
	2	–	4.7	93.8	–	1.5

**Table 3 materials-14-01455-t003:** The weight of the briquette used to determine the density.

Parameter	g
Weight of the briquette in the air	24.92
Mass of the briquette soaked in paraffin	25.08
Weight of the briquette in water	21.47

**Table 4 materials-14-01455-t004:** The results of hardness measurements on back surface in HLD.

The Surface Designation		BA	BB	BC	BD	BE	BF
top	1	–	183	252	281	208	–
	2	–	131	302	279	213	113
	3	–	126	225	218	193	127
	4	141	195	224	279	203	166
	5	139	107	235	220	165	119
	6	119	132	200	152	151	114
bottom	7	–	118	176	155	152	–

**Table 5 materials-14-01455-t005:** Results of hardness measurements on front surface in HLD.

The Surface Designation		FA	FB	FC	FD	FE	FF	FG	FH	FI
top	1	-	188	124	112	141	234	129	138	139
	2	171	142	155	171	167	147	142	125	165
	3	175	169	207	209	236	220	171	190	159
	4	258	178	212	245	230	197	176	172	147
	5	186	164	170	244	130	147	131	131	–
bottom	6	148	204	157	138	121	149	189	–	–

**Table 6 materials-14-01455-t006:** The results of surface roughness measurements on the back surface.

Parameter (μm)	The Area of the Measurement					
	BA4	BB4	BC4	BD4	BE4	BF4
R_a_	12.33	14.52	16.13	16.84	17.20	15.48
R_q_	15.16	18.37	19.99	21.14	21.56	18.95
R_t_	155.78	233.19	198.54	237.84	234.26	220.94

**Table 7 materials-14-01455-t007:** The results of surface roughness measurements on the front surface.

Parameter (μm)	The Area of the Measurement			
	FF1	FF2	FF3	FF4
R_a_	19.69	14.09	33.25	45.74
R_q_	25.86	17.70	42.34	58.39
R_t_	264.35	148.18	338.02	495.21

**Table 8 materials-14-01455-t008:** Average values of the porosity for individual areas of the sample.

Cross-Section	Porosity (%)					
	1	2	3	4	5	6
W	23.4	13.5	20.3	27.6	35.3	45.0
X	14.7	16.8	32.2	38.8	41.6	53.3
Y	-	28.4	37.8	41.1	47.5	-

## Data Availability

The data presented in this study are available on request from the corresponding author.

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
