# Peer review of "The Analysis of the Morphology of the Saddle-Shaped Bronze Chips Briquettes Produced in the Roller Press"

_materials, 2021, doi:10.3390/ma14061455_

Round 1

Reviewer 1 Report

Taking into account a general experimental procedure and analysis, the paper is well organized and full of experimental evidences how you can see.

The authors investigated the physical and mechanical properties of the briquettes and their external surfaces as well as on their cross-sections: density, chemical composition, microstructure analysis obtained with a 3D and scanning microscope, surface roughness and hardness. It is enough to be considered a good paper also with the minimum of the evaluation.

Only 3 minor aspects to revise:

  1. Since all the authors have the same affiliation name, they should use only number "1" for all of them;
  2. Figure 11: scale is missing;
  3. Figures 18-21 : error bar is missing.

Author Response

Dear Reviewer,

Thank you very much for taking the time to read our manuscript thoroughly and make recommendations for its correction and improvement. We have read the comments carefully and have responded to all your comments.

Remark 1

Since all the authors have the same affiliation name, they should use only number "1" for all of them

Our affiliation seems the same but our faculty is different.

Remark 2

Figure 11: scale is missing;

We added the scale to the figure.

Remark 3

Figures 18-21 : error bar is missing.

We have added bars with the standard error.

Reviewer 2 Report

This publication presents very nice and detailed experimental results, but conclusions of the work are not sufficient for the interest of readers. A better internal densification of the top part of the briquette is a logically expected, but descriptive conclusion: „It cannot be stated unequivocally, the roughness test results correlated with the hardness test results, i.e. the higher the hardness, the lower the surface roughness, because the correlations on edge 1 are different from that on edge 2.“ needs to be explained. The limiting condition of briquetting is the size of the chip and its mechanical properties. Of course, very hard and brittle materials cannot be successfully briquetted, because the chips break up during pressing and do not interconnect, ie the core of the material compressed from copper alloy chips and the core of the same copper alloy are not comparable! Therefore, this research is still "neglected". The relationship between the geometric properties of the surface and the physical properties is generally explained by "surface integrity" (by the residual stresses, hardness and microstructure of the material). The change in hardness in the surface layer is due to both mechanical action and thermal loading of the surface. In practice, there are three basic hardnesses the surface layer, namely in relation to the core of the material! (high surface hardness with decrease to core hardness, high surface hardness with decrease of hardness below core hardness with subsequent increase to core hardness, low surface hardness, which gradually increases to core hardness). The inquisitive reader will certainly ask: What is the bond between the surface and the core of the material in this case? Conclusions need to be supplemented to make it clear that, for example, follow-up research will continue in this direction.

Author Response

Dear Reviewer,

Thank you very much for taking the time to read our manuscript thoroughly and make recommendations for its correction and improvement. We have read the comments carefully and have responded to all your comments.

Remark 1

It cannot be stated unequivocally, the roughness test results correlated with the hardness test results, i.e. the higher the hardness, the lower the surface roughness, because the correlations on edge 1 are different from that on edge 2.“

We have changed it in the manuscript (page 17, sentence: For both the front and back surfaces, the highest surface roughness is obtained in the central part of the briquette.).

Remark 2

The change in hardness in the surface layer is due to both mechanical action and thermal loading of the surface. In practice, there are three basic hardnesses the surface layer, namely in relation to the core of the material! (high surface hardness with decrease to core hardness, high surface hardness with decrease of hardness below core hardness with subsequent increase to core hardness, low surface hardness, which gradually increases to core hardness).

The results of the measurements of hardness and roughness of the surface layer show only a partial correlation. This is due to many factors and also because the hardness was assessed by the Leeb dynamic method and then converted. This methodology was adopted due to the characteristics of the material in terms of the cohesion and geometry. Therefore, an increase in roughness in this case may act on the resilience (which was basically measured). The material under the surface layer, especially the degree of its compaction, also plays an important role. Conceptual work on the appropriate methodology for measuring hardness in the briquette cross-section is currently underway.

Remark 3

Conclusions need to be supplemented to make it clear that, for example, follow-up research will continue in this direction

We have added the following sentences in the manuscript: “Degree of compaction of the material under the surface layer as well as interaction between briquette core and its surface also plays an important role and will be investigated in the next stages of this research. Conceptual work on the appropriate methodology for measuring hardness in the briquette cross-section is currently underway.”.

Reviewer 3 Report

Dear authors,

I find your research interesting and well thought out, as well as important for the technological aspect. 

However, there are a couple of issues that I suggest you amend before the publication of the manuscript. In my opinion, the following points would make your manuscript better:

Introduction:

I personally found the introduction hard to read. It  should be broken up in a couple of shorter paragraphs, each adressing a specific problem/topic. The references are also hard to follow, becase same references appear numerous times throughout the manuscript. I suggest the authors cite a specific reference at the most appropriate place.

Experimental:

Generally gives suffficient information on the experimental work, all the steps that were carried out are explained in detail. Missing is sample preparation (I suppose it was just polished?) for the SEM/EDS work. Also, no details on EDS analysis were given.

Results and discussion:

General remark on the micrographs: the scale bars are not legible. Please make them bolder/larger/easier to read. Also, if you really want to show the EDS spectra, try to incomporate them in the Figure – no need to add an additional figure just for EDS spectrum, you already have a table showing results. And it is EDS spectrum, not X-ray diffraction pattern – this is an entirely different technique, not SEM-based, and it gives you complely different information.

Figure 12: can you make all the panels teh same size, it would be more appealing to the eye.

Figure 13 and Figure 14: they are essentially one figure. Mark both spots of EDS analysis, there is no need to show the spectra (if you want to highligh the differences, maybe plot them one on top of the other), and the table can be combined for both analyses.

Fig 18-Fig 21 can again be combined into a multipanel Figure, which would also make the comparison easier for the reader. What is the justification for the lines connecting the points? It is not mentioned in the text nor in the figure itself. Error bars of the measurements would also give some information, if possible please suppy them.

Conclusions: well written, summarise the results in a concise way.

Author Response

Dear Reviewer,

Thank you very much for reviewing our manuscript. According to the comments and the questions, we have carefully revised the article text. Below we answer to all the remarks. The changes in the manuscript have been highlighted in the tracking mode.

Remark 1

I personally found the introduction hard to read. It  should be broken up in a couple of shorter paragraphs, each addressing a specific problem/topic. The references are also hard to follow, because same references appear numerous times throughout the manuscript. I suggest the authors cite a specific reference at the most appropriate place.

We have reorganized the references.

Remark 2

Missing is sample preparation (I suppose it was just polished?) for the SEM/EDS work. Also, no details on EDS analysis were given.

We have added the details about sample preparation and EDS analysis to the manuscript.

Remark 3

the scale bars are not legible. Please make them bolder/larger/easier to read.

We have improved the scale bars.

Remark 4

And it is EDS spectrum, not X-ray diffraction pattern – this is an entirely different technique, not SEM-based, and it gives you complely different information.

We apologize for mistake in the name of EDS spots - we understand the difference regarding these two research techniques. We have corrected it in the manuscript.

Remark 5

Figure 12: can you make all the panels the same size, it would be more appealing to the eye.

We have corrected it in the manuscript.

Remark 6

Figure 13 and Figure 14: they are essentially one figure. Mark both spots of EDS analysis, there is no need to show the spectra (if you want to highligh the differences, maybe plot them one on top of the other), and the table can be combined for both analyses.

We changed it in the article.

Remark 7

Fig 18-Fig 21 can again be combined into a multipanel Figure, which would also make the comparison easier for the reader. What is the justification for the lines connecting the points? It is not mentioned in the text nor in the figure itself. Error bars of the measurements would also give some information, if possible please suppy them.

The lines in the figures (18-21 in the first version) were to show the general tendency of changes, therefore their character was changed to a broken line.

Reviewer 4 Report

The paper is poorly organized, and I suggest authors reshape the paper. It is a project report rather than a research paper. 

Author Response

Thank you for your feedback. The adopted work organization was to present as clearly as possible the applied research methodology, which in our opinion is very important, because it is to be used to conduct similar research for other briquetted materials. We admit, that focusing on research methodology can actually create the impression of expert work. In the case of further research (after this initiating work) and publications, we plan to change the organization of work so that it presents the scientific issues more clearly. It will be possible because we will be able to refer to this work to some extent in terms of the adopted research methodology.

Reviewer 5 Report

Recycling of industrial waste is of great importance. The study of the physical properties of compacted metal waste is important for their application in metallurgical production and for improving recycling technologies. The article contains useful information about the physical and mechanical properties of saddle-shaped copper alloy chips briquettes produced in a roller press. The results obtained are new and important in two directions, for the design of a roller press and the subsequent remelting of secondary raw materials. - Legends under the figures (10, 13-s, 14-s) would be changed. This is not (X-ray diffraction pattern) but Energy Dispersive Spectrum. Please, specify labels along the axes in eV. The spectrum should contain lines of carbon and oxygen, which are part of pollution. And in table 1 it is necessary to indicate the concentration of these uncharacteristic impurities. It is obvious that the compacted material is not absolutely pure. it is necessary to indicate the scale bar on microstructure 11, as is common. Figure 25 – “Pictures” would be changed to “microstructure”.

Author Response

Dear Reviewer,

Thank you very much for taking the time to read our manuscript thoroughly and make recommendations.

Remark  1

Legends under the figures (10, 13-s, 14-s) would be changed. This is not (X-ray diffraction pattern) but Energy Dispersive Spectrum.

We apologize for the mistake in the name of EDS spots - we understand the difference regarding these two research techniques. We have corrected it in the manuscript.

Remark  2

The spectrum should contain lines of carbon and oxygen, which are part of pollution.

The applied EDS research technique gives questionable quantitative results for low-energy elements (oxygen, carbon), therefore they are not included in the table

Remark 3

it is necessary to indicate the scale bar on microstructure 11,

We have corrected it in the manuscript.

Remark  4

Figure 25 – “Pictures” would be changed to “microstructure”.

We have corrected it in the manuscript.

Round 2

Reviewer 3 Report

Dear authors, 

the revised manuscripts flows much better now that you have incorporated some changes. All in all, the work you performed is interesting and best luck in your future work!

Reviewer 4 Report

NA